# SRGP-1/srGAP and AFD-1/afadin stabilize HMP-1/α-catenin at rosettes to seal internalization sites following gastrulation in *C. elegans*

**Joel M. Serre[1], Mark M. Slabodnick[2,3], Bob Goldstein[2], Jeff Hardin[1,4]***

**1** Program in Genetics University of Wisconsin-Madison, Wisconsin, United States of America, **2** Department of Biology, University of North Carolina at Chapel Hill, Chapel Hill, North Carolina, United States of America, **3** Department of Biology, Knox University, Galesburg, Illinois, United States of America, **4** Department of Integrative Biology, University of Wisconsin-Madison, Wisconsin, United States of America

* jdhardin@wisc.edu

**Data Availability Statement:** All relevant data are within the manuscript and its Supporting Information files.

## Abstract

A hallmark of gastrulation is the establishment of germ layers by internalization of cells initially on the exterior. In *C. elegans* the end of gastrulation is marked by the closure of the ventral cleft, a structure formed as cells internalize during gastrulation, and the subsequent rearrangement of adjacent neuroblasts that remain on the surface. We found that a nonsense allele of *srgp-1/srGAP* leads to 10–15% cleft closure failure. Deletion of the SRGP-1/srGAP C-terminal domain led to a comparable rate of cleft closure failure, whereas deletion of the N-terminal F-BAR region resulted in milder defects. Loss of the SRGP-1/srGAP C-terminus or F-BAR domain results in defects in rosette formation and defective clustering of HMP-1/α-catenin in surface cells during cleft closure. A mutant form of HMP-1/α-catenin with an open M domain can suppress cleft closure defects in *srgp-1* mutant backgrounds, suggesting that this mutation acts as a gain-of-function allele. Since SRGP-1 binding to HMP-1/α-catenin is not favored in this case, we sought another HMP-1 interactor that might be recruited when HMP-1/α-catenin is constitutively open. A good candidate is AFD-1/afadin, which genetically interacts with cadherin-based adhesion later during embryonic elongation. AFD-1/afadin is prominently expressed at the vertex of neuroblast rosettes in wildtype, and depletion of AFD-1/afadin increases cleft closure defects in *srgp-1/srGAP* and *hmp-1^{R551/554A}/α-catenin* backgrounds. We propose that SRGP-1/srGAP promotes nascent junction formation in rosettes; as junctions mature and sustain higher levels of tension, the M domain of HMP-1/α-catenin opens, allowing maturing junctions to transition from recruitment of SRGP-1/srGAP to AFD-1/afadin. Our work identifies new roles for α-catenin interactors during a process crucial to metazoan development.

## Author summary

A key feature of early embryonic development in animals is gastrulation. Gastrulation often involves the movement of cells initially on the surface to the interior, where they

**Funding:** JS and JH were supported by NIH grants R01GM058038, R01GM127687, and R35GM145312. BG was supported by NIH grant R35GM134838. MMS was supported by NIH grant F32GM119348. The funders had no role in study design, data collection and analysis, decision to publish, or preparation of the manuscript.

**Competing interests:** The authors have declared that no competing interests exist.

serve as founder cells for the formation of key tissues in the embryo. We used the early embryo of the nematode, *C. elegans*, as a model system to investigate how cells on the surface seal gaps left behind as cells move into the interior. We found that two parts of a protein known as SRGP-1/srGAP are important for the formation of "rosettes" at the end of gastrulation: a part that binds membranes and another that binds a key adhesion protein known as HMP-1/α-catenin. Defects in SRGP-1 function lead to defects in the clustering of α-catenin that are important for strengthening cell-cell adhesion. These defects can be bypassed by activating mutations in α-catenin. We found that a second protein, AFD-1/afadin, is normally prominently expressed at rosettes, but that depletion of AFD-1/afadin frequently leads to sealing defects. SRGP-1 and AFD-1 appear to act largely in parallel during sealing. Our work identifies new roles for α-catenin interactors during a process crucial to animal development.

## Introduction

Gastrulation is a hallmark of metazoan development that establishes the basic body plan [1]. In many organisms, internalization of founder cells that form the three primary germ layers, as well as primordial germ cells, occurs via detachment of the apical surfaces of individual cells from the embryo's exterior [2–5]. Such internalization can involve an epithelial-mesenchymal transition (EMT), as cells dismantle their cell-cell junctional machinery and detach [6,7]; in other cases, a true EMT does not occur [5,8]. Neighboring cells that remain on the exterior must seal the breach left behind by internalizing cells, rearranging and making new cell-cell junctional connections as they do so. While cell internalization is essential for successful gastrulation in numerous organisms, most of the focus thus far has been on cellular events within internalizing cells; relatively less attention has been paid to neighboring cells that seal the embryonic exterior.

The early *C. elegans* embryo is a useful model system for understanding changes in cell-cell adhesion associated with cell internalization. Gastrulation in *C. elegans* involves stereotypical events on the ventral surface of the embryo that internalize endodermal, mesodermal, and germ cell precursors [5,9]. The best studied of these events is the internalization of Ea and Ep, the endodermal precursors whose internalization marks the beginning of gastrulation. Ea/p undergo myosin-mediated apical constriction [8,10–13]. Germ cell precursors rely on a different mechanism, involving cadherin-dependent "hitchhiking" [14]. Concomitant with internalization of Ea/p, neighboring cells have been observed to produced protrusions that may aid resealing of the embryo's surface via active crawling [15]. Together with apical constriction of internalizing cells themselves, these movements are thought to aid cell internalization and simultaneous resealing of the ventral surface [12,15].

A ventral cleft forms on the surface of the embryo as the last sets of cells are internalized at the end of gastrulation. The ventral cleft is surrounded by neuroblasts derived from ABplp and ABprp in the posterior and ABalp and ABarp in the anterior. The ventral gastrulation cleft is subsequently closed via movements of ventral neuroblasts toward the ventral midline between 230 and 290 minutes postfertilization, around 120 minutes after gastrulation begins (Fig 1A), causing the ventral cleft to disappear approximately one hour before the movements of ventral epidermal enclosure begin [16,17]. Failures in ventral cleft closure lead to highly penetrant failure of ventral enclosure (for reviews of this process, see [9,18,19]).

Defects in the movement of neuroblasts to close the ventral cleft are observed in embryos defective in several cell signaling pathways, including those involving Eph/ephrin signaling

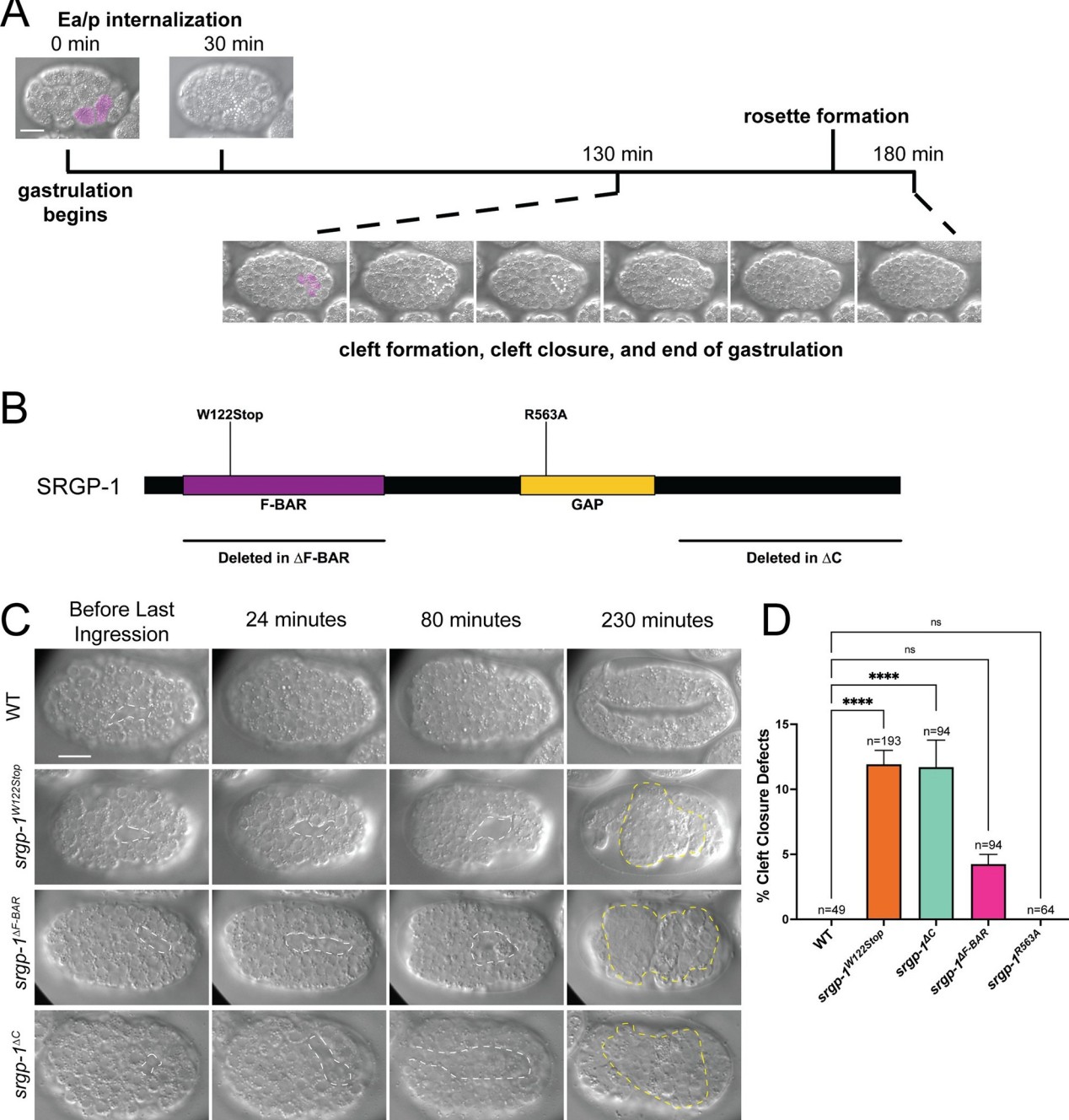

**Fig 1. Genetic perturbation of *srgp-1* leads to cleft closure defects. (A)** Summary timeline of events relevant to this study at 20˚C. The beginning of gastrulation in *C. elegans* is marked by the internalization of the endodermal precursors Ea/p. Ventral cleft formation and closure begin approximately 100 min. later. Internalizing cells are indicated by pink pseudo-coloring. **(B)** A map depicting the domains of SRGP-1 including mutants used in this study (*srgp-1^{W122Stop}*, *srgp-1^{R563A}*, *srgp-1^{ΔF-BAR}*, and *srgp-1^{ΔC}*). **(C)** DIC images of embryos over 230 minutes. White dotted lines depict the ventral cleft. The first row depicts a typical wild-type embryo proceeding through cleft closure and into early elongation. *srgp-1^{W122Stop}*, *srgp-1^{ΔF-BAR}*, and *srgp-1^{ΔC}* mutants all display cleft closure failure, resulting in extruded endoderm (yellow dotted lines). Scale bar is 10 μm. **(D)** A graph depicting percentage cleft closure defects in wild-type and *srgp-1* mutants. \*\*\*\*, p < 0.0001; \*, p < 0.05.

[16,20,21], PTP-3/LAR (Leukocyte Common Antigen Related Receptor, a protein tyrosine phosphatase; [22]), semaphorin-2A/MAB-20/plexin signaling [23,24], and the *C. elegans* Kallmann syndrome ortholog *kal-1* [25,26]. Such defects result in an enlarged or persistent ventral cleft; if the ventral cleft is not closed by the time of epidermal enclosure, enclosure movements are often disrupted.

The motile events downstream of cell signaling at the ventral cleft are poorly understood; loss of function of the SCAR/WAVE gene *wve-1* leads to significant defects in ventral neuroblast organization [27], suggesting that actin-based motility may be important for ventral neuroblast movement. Filopodial protrusions have been observed during cleft closure, but their significance is unclear [28]. As ventral neuroblasts move together, surrounding cells adjacent to the cleft must rearrange as the cleft closes [12]. After the events of ventral cleft closure, the neuroblasts that seal the cleft divide and rearrange to form part of the presumptive ventral nerve cord before the embryo begins to elongate into a vermiform shape [9,29]. Ventral neuroblasts later accumulate myosin foci and cadherin complex proteins [30].

Internalization of cells during gastrulation in *C. elegans* involves detachment of cells from their neighbors and establishment of new connections among cells remaining at the ventral surface, so changes in cell-cell adhesion must presumably occur during this process. The *C. elegans* cadherin/catenin complex (CCC) has been the focus of significant attention in this regard. The core components of the CCC, HMR-1/cadherin, HMP-2/β-catenin, and HMP-1/α-catenin, are present in the early embryo before gastrulation begins [11,14,31]. While there is not an essential requirement for cadherin-dependent adhesion during Ea/p internalization in otherwise wild-type embryos, there is a synergistic requirement for the cadherin complex when the L1CAM homologue SAX-7 or CED-5/DOCK180 is depleted [32,33]. Accumulation of CCC components at the interface between cells that internalize and those that remain on the surface has been proposed to aid recruitment of actomyosin contractile networks necessary for internalization [5,11], after engagement of an actomyosin-mediated "clutch" in Ea/p [13].

Much of the focus regarding the CCC during gastrulation has been on the internalizing cells, specifically Ea/p. Requirements for the CCC in subsequent internalizations have not been specifically analyzed, nor has the role of the CCC in resealing the ventral surface after internalization been assessed. We set out to investigate roles for the core CCC component, HMP-1/α-catenin, in these processes. In addition, we turned our attention to SRGP-1/srGAP, the lone slit/robo GTPase activating protein in *C. elegans* [28,34]. We showed previously that SRGP-1 is a modulator of cell-cell adhesion during the later events of ventral enclosure [28] and embryonic elongation [35]. In addition, however, *srgp-1* knockdown in *hmp-1(fe4)/α-catenin* mutants leads to Gex (Gut on the exterior) phenotypes due to a failure to complete cleft closure [28], implicating it in ventral cleft sealing at the end of gastrulation.

SRGP-1/srGAP is a homolog of vertebrate Slit/Robo GTPase Activating Proteins (srGAPs), which have an N-terminal F-BAR domain that associates with curved membranes, a central RhoGAP domain, and an SH3 domain which has been shown to associate with various other factors such as WAVE, WASP, and Lamellipodin [36–38]. SRGP-1 in *C. elegans* does not contain an SH3 domain; nevertheless, we showed previously that the SRGP-1 C-terminus interacts with both the N-terminal half of SRGP-1 [28] and with HMP-1/α-catenin [35]. Overexpression of the F-BAR domain of SRGP-1 leads to ectopic membrane tubulations. The C-terminus of SRGP-1 is required to recruit HMP-1/α-catenin into these tubulations [28] and for normal HMP-1/α-catenin dynamics [35], consistent with a role for the SRGP-1 C terminus in coordinating the interaction with HMP-1/α-catenin.

We also showed recently that the C terminus of SRGP-1/srGAP physically binds to the middle (M) domain of HMP-1/α-catenin. Surprisingly, unlike the interaction of vertebrate vinculin with the αE-catenin M domain, in which maximal vinculin binding to αE-catenin is

favored by complete unfurling of the αE-catenin M domain [39] and in which salt bridge mutations in the αE-catenin M domain are predicted to activate it for vinculin binding [40–44], the introduction of destabilizing mutations designed to disrupt key salt bridges in the HMP-1 M domain abrogate the interaction with the SRGP-1 C terminus [35]. This suggests that SRGP-1 interacts with conformations of HMP-1/α-catenin in which the M domain is not fully extended, and further raises the question as to whether other HMP-1/α-catnein effectors interact with different conformational states of HMP-1.

Here we investigated the role of SRGP-1/srGAP prior to epidermal morphogenesis, as the ventral surface seals the final breaches due to cell internalization at the end of gastrulation. We found that SRGP-1 is required for normal cell behavior, cell morphology, and HMP-1/α-catenin recruitment during this essential process. We also found that destabilizing salt bridge mutations within the M (middle) domain of HMP-1/α-catenin, which cause the M domain to remain in an extended state and abrogate binding by the SRGP-1 C terminus [35], are able to suppress SRGP-1 phenotypes. This suppression may be in part due to increased recruitment of components that interact with an open conformation of the M domain, including the *C. elegans* afadin homologue, AFD-1.

## Results

### Mutations in srgp-1/srGAP lead to cleft closure defects

Our prior work established a role for SRGP-1/srGAP in the embryonic epidermis in *C. elegans* [28,35], but srGAPs in vertebrates were originally identified through their roles in the developing nervous system [38,45–48]. In *C. elegans*, a majority of neuroblasts are found on the ventral side of the embryo following gastrulation [9,17]. These neuroblasts must (1) adhere to one another to keep other tissues internalized during gastrulation (reviewed in [5,9]), (2) divide and rearrange to form part of the ventral nerve cord [29], and (3) act as a substrate for the epidermis, which undergoes epiboly during ventral enclosure [23,30,49]. Using 4D DIC microscopy, we observed that an appreciable percentage (11.1%) of homozygotes for *srgp-1*(*jc72*), a nonsense allele that functions as a null hereafter referred to as *srgp-1*[W122Stop] (Fig 1B), do not complete ventral cleft closure at the end of gastrulation, leading to endodermal precursors being extruded when the epidermis attempts to undergo epiboly and the contractions normally associated with embryonic elongation (Fig 1C, second row).

SRGP-1/srGAP has three major functional domains: (1) an N-terminal F-BAR domain, (2) a central GAP domain, and (3) an unstructured C-terminal region that is involved in protein-protein interactions (see Fig 1B, [28,34,35,50]}) We explored whether one of these domains might be important for SRGP-1 function during cleft closure. Using CRISPR/Cas9 methodology, we generated the following alleles: *srgp-1*[ΔF-BAR], missense allele *srgp-1*[R563A], which prevents GAP activity [34,51], and *srgp-1*[ΔC], which deletes most of the region C-terminal to the GAP domain. Loss of the SRGP-1 F-BAR domain and C-terminal region both led to cleft closure defects following gastrulation (Fig 1C). The percentages of embryos that displayed cleft closure defects were similar between *srgp-1*[ΔC] and *srgp-1*[W122Stop] alleles; while we observed cleft closure defects in *srgp-1*[ΔF-BAR] mutant embryos and did not observe any cleft closure defects in wild-type embryos, the lower frequency observed in the *srgp-1*[ΔF-BAR] background did not rise to the level of statistical significance compared to wildtype (Fig 1D). As in our previous studies examining the epidermal functions of SRGP-1 [28], we did not observe any obvious defects in embryos lacking SRGP-1 GAP functionality. These results suggest that important aspects of SRGP-1 function during cleft closure are mediated through its C terminus, with the F-BAR domain playing a supporting role.

## HMP-1/α-catenin and SRGP-1/ssrGAP co-localize at the vertices of rosettes following the last internalization events of gastrulation

Gastrulation in *C. elegans* involves the internalization of progenitor cells that generate endoderm, mesoderm, and germline tissues. As these cells move into the interior, they undergo apical constriction. As they do so, neighboring cells form transient rosettes to cover the space vacated by the departing cells [12]. Such rosettes during *C. elegans* gastrulation are superficially similar to the rosettes that are a prominent feature during some cases of convergent extension, such as germband extension in the *Drosophila* embryo [52–54], but involve very different cellular events. The former involve a fundamentally different process (apical constriction of internalizing cells); the latter are driven by edge contraction within a contiguous epithelium. Whereas rosette formation during convergent extension has been documented in *C. elegans* [29], the adhesion events associated with rosette formation during cell sealing have not been investigated. We therefore examined endogenously tagged HMP-1/α-catenin::mScarlet-I and SRGP-1/srGAP::mNeonGreen in living embryos, beginning with ventral cleft formation through the final internalization events of gastrulation, which occur after cleft closure (Fig 2A).

Two rosettes form and resolve at this stage, involving cells born on the left and right sides of the ventral cleft (Fig 2A, yellow dotted line; B, colored cells). At the vertex of the anterior rosette, where cells internalize, we observed a bright accumulation of HMP-1/α-catenin:: mScarlet-I immediately after the internalization event (Fig 2A, white arrowhead). Subsequent to the accumulation of HMP-1/α-catenin in the anterior rosette, the posterior rosette resolved and elongated along the anterior-posterior axis, forming new cell contacts as it did so (Fig 2A; yellow arrowheads at 10 min indicate direction of cell movement by 20 min). Significantly, SRGP-1 also accumulated at vertices in both the anterior and posterior rosettes (Fig 2A, 0 min, arrowheads).

We previously demonstrated that homozygotes carrying a nonsense allele of *srgp-1* display decreased HMP-1/α-catenin junctional intensity [35]. We therefore sought to determine whether *srgp-1* mutant backgrounds could influence the localization of HMP-1 during rosette formation, focusing on the anterior rosette. We examined localization of HMP-1::mScarlet-I within the anterior rosette before and after the final set of cell internalizations (Fig 3; blue dotted line indicates internalizing cells). In a full-length, endogenously tagged *srgp-1* background, HMP-1 accumulated at the vertex formed by the disappearance of internalizing cells (Fig 3A, 0 min). In contrast, in *srgp-1*$^{ΔF-BAR}$ mutants HMP-1 at the vertex failed to coalesce into a single cluster (Fig 3B, 0 minutes). In addition, a stable rosette no longer formed, and the remaining neuroblasts instead coalesced into two rows with no central vertex (Fig 3B, yellow lines). In *srgp-1*$^{ΔC}$ mutants clusters of SRGP-1 accumulated within neuroblasts with no apparent pattern; while HMP-1 was still able to coalesce around the rosette, multiple clusters with accumulated HMP-1 were visible (Fig 3C). Using the basic phenotypic classes depicted in Fig 3 ("focused" as in Fig 3A, "scattered" as in Fig 3B, and "unfocused" as in Fig 3C), we performed semi-quantitiave analysis of wild-type and *srgp-1* mutant embryos (S1 Fig). Wild-type embryos displayed focused HMP-1/α-catenin at the vertex in 9 of 9 embryos observed. In 7 of 8 *srgp-1*$^{ΔF-BAR}$ embryos imaged, HMP-1::mScarlet-I appeared scattered, as in the embryo shown in Fig 3B; in one instance HMP-1 appeared unfocused at the vertex in a manner similar to the embryo shown in Fig 3C. In 5 of 7 *srgp-1*$^{ΔC}$ embryos HMP-1 failed to focus at the vertex similar to the embryo shown in Fig 3C; in the remaining 2 embryos localization appeared focused, and superficially wild-type. The results for the three genotypes are significantly different (p < 0.0001, Chi-square test). Taken together, these results indicate that both the SRGP-1 N- and C-terminal regions have important roles during cleft closure. The F-BAR domain appears

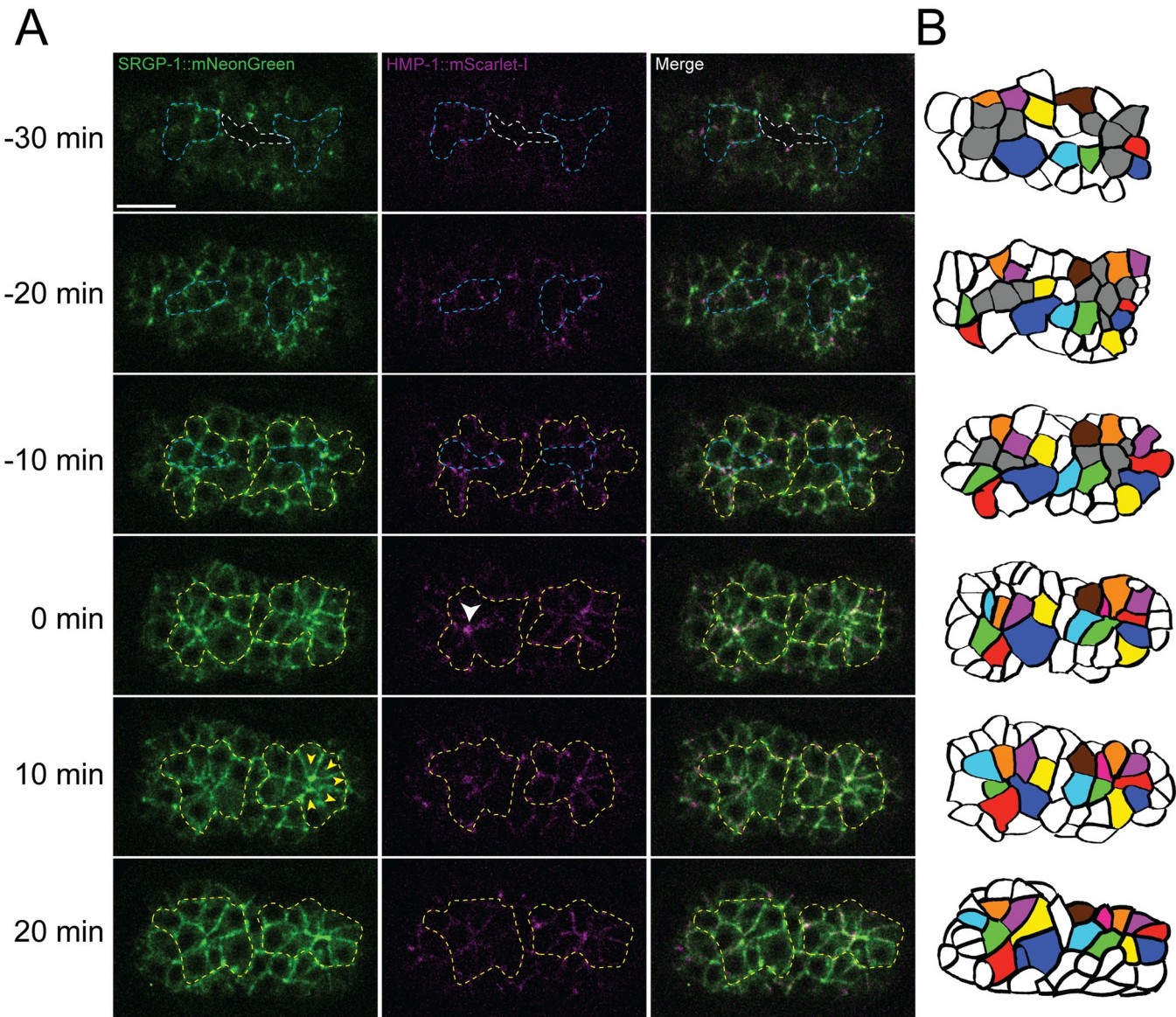

**Fig 2. Rosette formation leads to cell rearrangement during cleft closure. (A)** An embryo expressing SRGP-1::mNeonGreen and HMP-1::mScarlet-I before, during, and after anterior rosette formation (ventral view), showing the cell rearrangements that take place to seal the ventral cleft. Times are relative to formation of the anterior rosette (t = 0 min). Yellow dotted lines outline cells involved in rosette formation. Blue dotted lines indicate cells that internalize. White dotted lines indicate the ventral cleft. White arrowhead indicates the vertex of the anterior rosette. Yellow arrows indicate the direction of cell movement at the posterior end of the embryo following anterior rosette formation. **(B)** Cell tracings of the embryos in (A). Internalizing cells are colored grey, cells that form rosettes are colored, all other cells are white. Scale bar is 10 μm.

to be important for organizing the tips of cells within rosettes into a single vertex, whereas the C-terminus may play an important role in spatially organizing and interacting with HMP-1 during this process.

## Destabilizing the M domain of HMP-1/α-catenin suppresses cleft closure defects in srgp-1/srGAP mutants

The experiments in the previous section indicate that SRGP-1/srGAP subdomains are differentially required for fully wild-type HMP-1/α-catenin localization. We next sought to

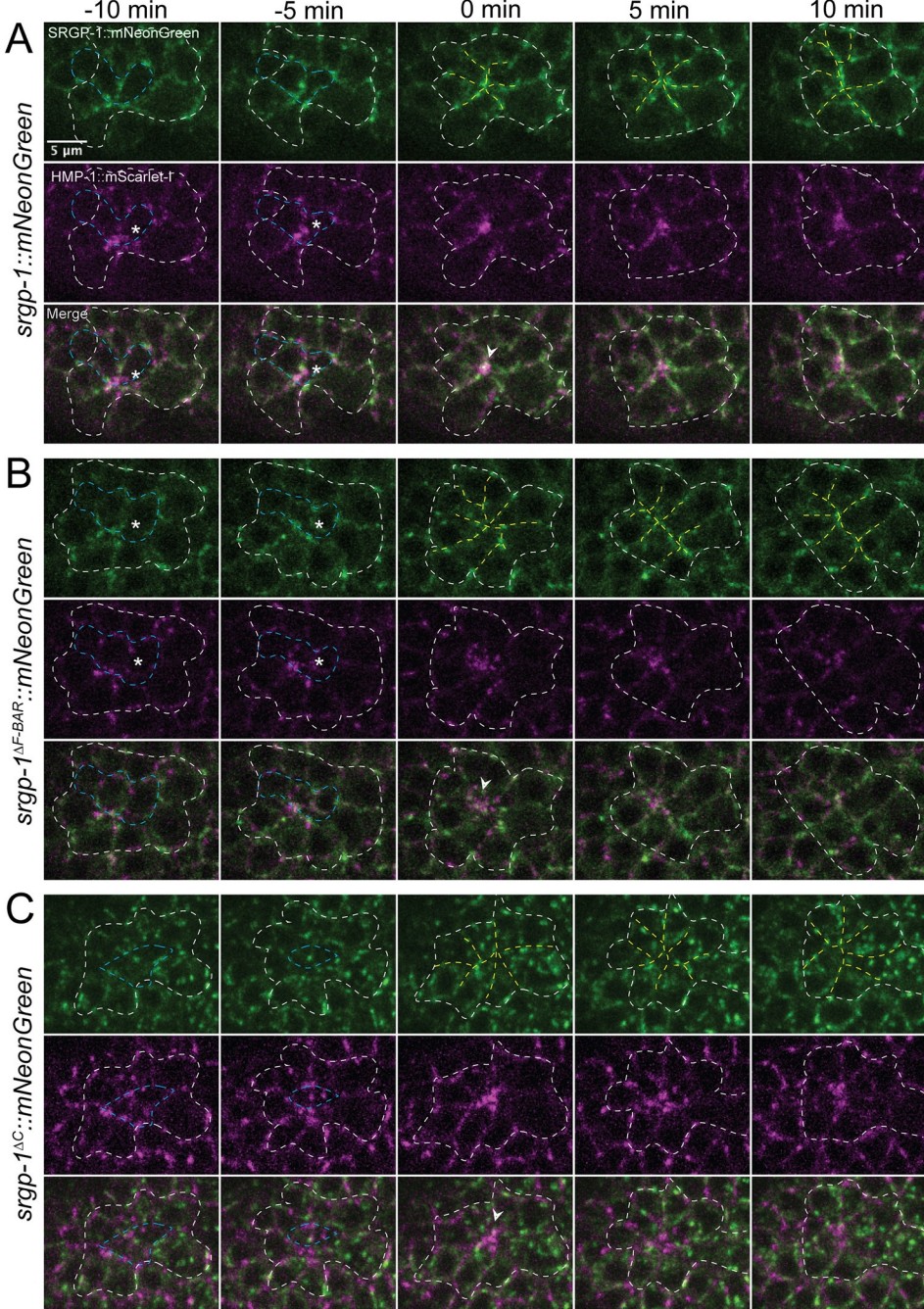

**Fig 3. *srgp-1* mutants display aberrant HMP-1::mScarlet-I aggregation and anterior rosette formation.**
Representative embryos expressing full-length or deleted SRGP-1::mNeonGreen, as well as HMP-1::mScarlet-I. (**A**) SRGP-1::mNeonGreen; (**B**) SRGP-1$^{\Delta F\text{-}BAR}$::mNeonGreen; (**C**) SRGP-1$^{\Delta C}$::mNeonGreen. Blue dotted lines indicate cell internalization that precedes formation of rosettes. White arrowheads indicate the vertex of the rosette following cell internalization. White dotted lines outline cells that contribute to the rosette; yellow dotted lines depict cells arranged around the vertex of the rosette. Scale bar is 5 μm.

determine if an activating mutation in HMP-1 could override the requirement for *srgp-1*. The M domain of HMP-1/α-catenin forms a closed structure that is stabilized by multiple salt bridges (Fig 4A; [55]). Mutating Arginines 551 and 554 to alanines prevents two of these salt

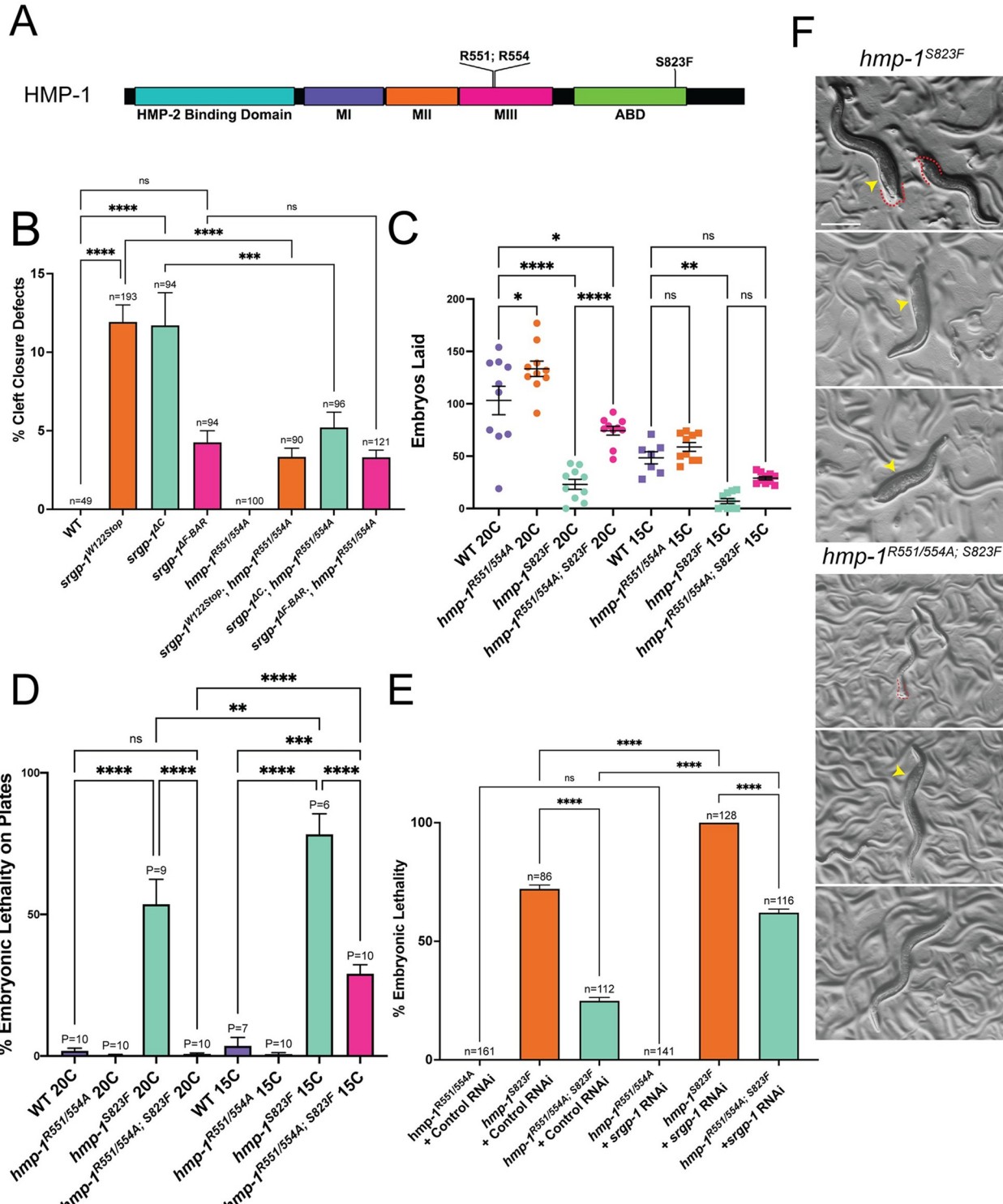

**Fig 4. The *hmp-1^R551/554A* mutation suppresses defects due to *srgp-1* loss of function and reduced actin binding ability of HMP-1. (A)** A domain map of HMP-1 depicting the sites of the R551/554A and S823F variations. **(B)** Percentage cleft closure defects in *srgp-1* and *hmp-1^R551/554A* mutants. **(C)** Fecundity of wildtype and various *hmp-1* mutants at 20˚C and 15˚C. **(D)** Embryonic lethality of wildtype and various *hmp-1* mutants at 20˚C and 15˚C. **(E)** Embryonic lethality of wildtype and various *hmp-1* mutants subjected to control (empty vector L4440) or *srgp-1* RNAi. **(F)** *hmp-1^S823F* and *hmp-1^R551/554A; S823F* hermaphrodites. Yellow arrowheads indicate abnormally shaped or swollen regions along the body. Red dotted lines indicate clubbed tails. Scale bar is 200 μm. ****, p < 0.0001; ***, p<0.001; **, p<0.01; *, p < 0.05.

bridges from forming between MII and MIII; as a result, the HMP-1 M domain adopts a constitutively open conformation that prevents the recruitment of the SRGP-1/srGAP C-terminus [35]. Although the *hmp-1*$^{R551/554A}$ mutation abrogates interaction with the C-terminus of SRGP-1, there is evidence in vertebrates that an extended conformation of α-catenin may activate actin binding and/or recruitment of other binding partners, including vinculin [39,42,44,56–58] and afadin [59]. We therefore assessed whether a constitutively open conformation of HMP-1 could bypass the requirement for SRGP-1 at the end of gastrulation.

Mutants homozygous for the *hmp-1/α-catenin* allele *hmp-1*$^{R551/554A}$ and either the *srgp-1*$^{W122Stop}$ or the *srpg-1*$^{ΔC}$ allele displayed fewer cleft closure defects compared to *srgp-1*$^{W122Stop}$ or *srpg-1*$^{ΔC}$ homozygotes with wild-type *hmp-1*; in contrast, there was no detectable change in frequency of cleft closure defects in *srgp-1*$^{ΔF-BAR}$ homozygotes when the salt bridge mutations were introduced (Fig 4B). These results suggest that an open conformation of the HMP-1 M domain can bypass some functions normally performed by the SRGP-1 C-terminu**s**. Since we were unable to detect any suppression in *srgp-1*$^{ΔF-BAR}$ homozygotes, the SRGP-1 F-BAR domain may still be required, presumably independently of binding of the SRGP-1 C terminus to the HMP-1 M domain.

To test if the salt bridge mutations in HMP-1/α-catenin truly behave as gain-of-function mutations, we introduced these mutations into the *hmp-1(fe4)* background. The *hmp-1(fe4)* allele, hereafter referred to as *hmp-1*$^{S823F}$, replaces a serine with a phenylalanine (S823F) within the actin binding domain of HMP-1 and behaves as a hypomorph (Fig 4A; [60,61]). *hmp-1*$^{S823F}$ homozygotes display morphogenetic failure and developmental arrest at a variety of stages, including during cleft closure. Using CRISPR/Cas9 we introduced the R551/554A mutations into the *hmp-1*$^{S823F}$ background. Hermaphrodites homozygous for *hmp-1*$^{R551/554A;}$ $^{S823F}$ on average laid more embryos and had minimal body morphology defects compared to *hmp-1*$^{S823F}$ homozygotes (Fig 4C and 4F). *hmp-1*$^{R551/554A; S823F}$ homozygous embryos also showed reduced lethality compared to embryos homozygous for *hmp-1*$^{S823F}$ (Fig 4D and 4E). We also observed that *hmp-1*$^{S823F}$ homozygotes exhibited increased embryonic lethality at colder temperatures (Fig 4D). We examined whether the *hmp-1*$^{R551/554A}$ mutation suppressed *hmp-1*$^{S823F}$ phenotypes at 15˚C. The introduction of the salt bridge mutations partially suppressed embryonic lethality at 15˚C, and, although it did not rise to the level of statistical significance, *hmp-1*$^{R551/554A;S823F}$ homozygotes reared at 15˚C had an increase in brood size compared to *hmp-1*$^{S823F}$ homozygotes (Fig 4C and 4D). We also noticed that *hmp-1*$^{S823F}$ embryos were very sensitive to mechanical pressure in 10% agar mounts and required mounting on 5% agar pads (S2 Fig), which we used for subsequent experiments utilizing this allele. *hmp-1*$^{R551/554A; S823F}$ embryos were less sensitive to mechanical pressure in 10% agar mounts, but fragility was not fully rescued by introducing the salt bridge mutation (S2 Fig). Taken together, these results indicate that the *hmp-1*$^{R551/554A}$ mutation acts as a gain-of-function allele that can partially offset reduction of tissue fragility resulting from the decreased actin binding activity caused by the C-terminal *S823F* mutation. Although we did not follow up this result further here, the increase in egg laying in *hmp-1*$^{R551/554A}$ mutants is interesting, and suggests that the *hmp-1*$^{R551/554A}$ mutations leads to a change in mechanical properties in the spermatheca, which displays actomyosin contractility [62,63] and dramatic junctional distension [64]. Significantly, SRGP-1 is expressed prominently there [28].

We previously utilized homozygotes for the *hmp-1/α-catenin* hypomorphic allele, *hmp-1 (fe4)* (hereafter *hmp-1*$^{S823F}$), as a sensitized background to identify modulators of cadherin-dependent adhesion [65] and identified *srgp-1/srGAP* as a strong enhancer of embryonic lethality in the *hmp-1*$^{S823F}$ background. RNAi knockdown of *srgp-1* resulted in nearly total embryonic lethality, while *srgp-1* knockdown by feeding RNAi in wild-type embryos had minimal effects. At least some of the synergistic lethality was caused by Gex phenotypes during

ventral enclosure [28]. We therefore examined the synergistic effect of *srgp-1* RNAi knockdown in *hmp-1*$^{R551/554A; S823F}$ embryos. The salt bridge mutation was able to reduce the embryonic lethality of *srgp-1(RNAi);hmp-1*$^{S823F}$ embryos significantly (Fig 4E). These results confirm that an open conformation of the HMP-1 M domain is able to bypass some requirements for SRGP-1, in addition to its ability to compensate for reduced actin binding activity mediated by the HMP-1 C terminus.

## Loss of afd-1/afadin function leads to increased frequency of cleft closure defects

Conformational changes within the vertebrate α-catenin M domain can affect its ability to recruit components that modulate cell adhesion. One such modulator in vertebrates is vinculin; when the αE-catenin M domain is extended, either via direct mechanical distension [66–68] or by introducing salt bridge mutations [58], its binding affinity for vinculin is increased. However, DEB-1, the vinculin homolog in *C. elegans*, does not interact with HMP-1 [55], and its expression is confined to muscle cells during development [69–71], ruling it out as a candidate HMP-1 interactor that could be positively affected by the *R551/554A* salt bridge mutations in the context of cleft closure. Another candidate modulator is AFD-1/afadin. Vertebrate afadin can bind to αE-catenin [72,73]. The *Drosophila* afadin, Canoe, localizes to cell-cell junctions and modulates morphogenesis in a variety of contexts in *Drosophila* [74–79], including rosette formation in the enxtending germband [43,80]. We showed previously that AFD-1/afadin can be co-immunoprecipitated with HMP-1 [81] (a result recently confirmed in another context [82]) and that loss of *afd-1* function synergizes with the *hmp-1*$^{S823F}$ mutation during later morphogenesis [65]. We therefore examined whether *afd-1* loss of function showed genetic interactions with *srgp-1/srGAP* and with *hmp-1/α-catenin* salt bridge mutations.

We first determined whether *afd-1/afadin* RNAi led to cleft closure defects and lethality in wild-type embryos or in embryos homozygous for the *hmp-1*$^{S823F}$ allele (Fig 5A–5C). Knockdown of *srgp-1/srGAP* or *afd-1* led to comparable levels of lethality and cleft closure defects in otherwise wild-type embryos on agar mounts. Moreover, RNAi against either *srgp-1* and *afd-1* caused a significant increase in cleft closure defects in *hmp-1*$^{S823F}$ embryos on plates or agar mounts (Fig 5A–5C). Since both *srgp-1* and *afd-1* knockdown increased the frequency of cleft closure defects in the *hmp-1/α-catenin* hypomorph, *hmp-1*$^{S823F}$, we examined how *afd-1* knockdown genetically interacted with *srgp-1* loss of function and with the *hmp-1*$^{R551/554A}$ mutation. In all backgrounds, *afd-1* knockdown resulted in an increase in cleft closure defects (Fig 5D; p-values for all pairwise comparisons in Fig 5D can be found in S3 Table). Significantly, *afd-1* knockdown in the *hmp-1*$^{R551/554A}$ background resulted in increased frequency of cleft closure defects compared to wildtype with *afd-1* knockdown, indicating that the salt bridge mutation is unable to bypass loss of *afd-1*. In *srgp-1*$^{W122Stop}$; *hmp-1*$^{R551/554A}$ double mutants, loss of *afd-1* resulted in a higher frequency of cleft closure defects than in *hmp-1*$^{R551/554A}$ alone, but less than in *srgp-1*$^{W122Stop}$ mutants alone, i.e. there is still partial rescue of loss of *srgp-1* function. This suggests that there may be effectors in addition to AFD-1 that act when HMP-1 adopts an open conformation, thereby enabling it to suppress cleft closure defects in the *srgp-1*$^{W122Stop}$ background.

Computational work was previously used to engineer the actin binding domain of human αE-catenin to bind actin with higher affinity [42]. Using protein alignment, we identified the homologous amino acids in HMP-1/α-catenin and generated *hmp-1*$^{QNLM676-679GSGS}$, which is predicted to bind actin with higher affinity (S3A and S3B Fig). Embryos homozygous for *hmp-1*$^{QNLM676-679GSGS}$ have a low level of embryonic lethality, which causes developmental arrest at various embryonic stages, including during cleft closure. These phenotypes were suppressed by loss of function of *srgp-1/srGAP* or *afd-1/afadin* (S3C and S3D Fig). These results suggest

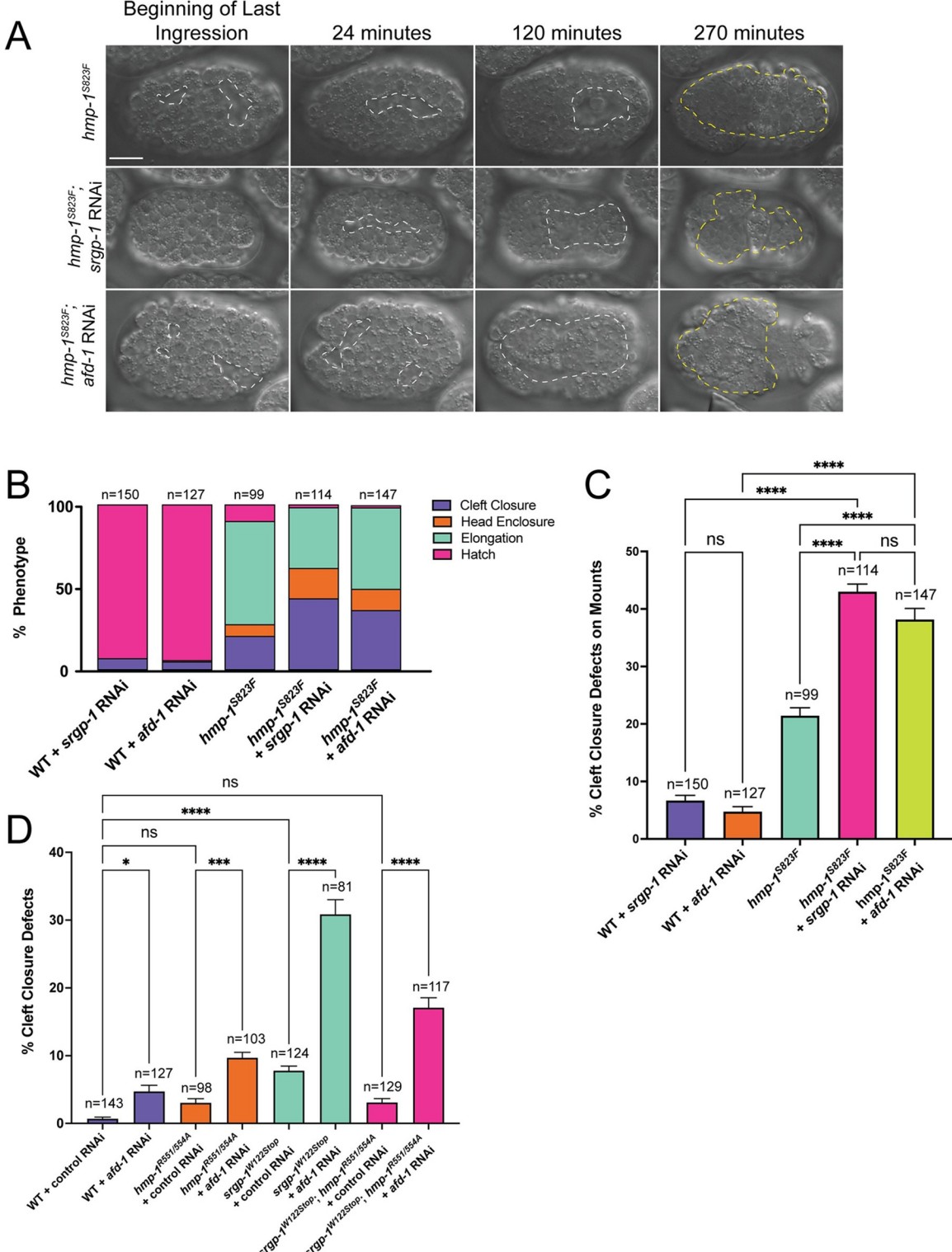

**Fig 5. *srgp-1* and *hmp-1* genetically interact with *afd-1* during cleft closure.** (**A**) DIC images of *hmp-1*$^{S823F}$, *hmp-1*$^{S823F}$; *srgp-1(RNAi)*, and *hmp-1*$^{S823F}$; *afd-1(RNAi)* embryos over the course of 270 minutes. Dotted white lines outline the ventral cleft. Yellow dotted lines indicate extruded gut. Scale bar is 10 μm. (**B**) A stacked bar plot indicating the percentage of embryos that die during cleft closure, head enclosure, and elongation in various genetic backgrounds. *afd-1* and *srgp-1* knockdown in the *hmp-1*$^{S823F}$ background both significantly increase the percentage of cleft closure defects. (**C**) Percent cleft closure defects in various genetic backgrounds with or without depletion of *srgp-1* and *afd-1* via RNAi. (**D**) Percentage of cleft closure defects in embryos treated with Control or *afd-1* RNAi. ****, p < 0.0001; ***, p<0.001; *, p < 0.05.

that HMP-1 stability and linkage to the actin network must be maintained within a dynamic range during processes that contribute to cleft closure.

## AFD-1/afadin localizes to the vertex of the anterior rosette at the end of gastrulation

Since genetic perturbation of *afd-1/afadin* had consequences for cleft closure, we next assessed the localization of AFD-1 during cleft closure. We visualized mKate2::AFD-1 and HMP-1/α-catenin:: GFP from cleft closure through rosette formation. While expression of mKate2::AFD-1 in the ventral neuroblasts was weak, precluding highly time-resolved, detailed 4d analysis of AFD-1 localization, we observed strong accumulation of AFD-1 at the vertex of the anterior rosette immediately following the final internalization events of gastrulation, which quickly dispersed as the rosette resolved (Fig 6A). We next examined HMP-1/α-catenin accumulation and localization at the vertex of the anterior rosette in *afd-1* loss-of-function backgrounds, combined with mutations in *srgp-1/srGAP* and *hmp-1/α-catenin* we had previously analyzed. (Fig 6B–6D). Total accumulation of HMP-1/α-catenin::GFP increased in the *hmp-1$^{R551/554A}$* mutant background compared to wild-type, consistent with constitutive activation of HMP-1. Moreover, the additional recruitment of AFD-1 in this background suggests that AFD-1 is a HMP-1 effector even when it is extended. Thus depleting embryos of AFD-1 might be expected to have effects on HMP-1 that cannot be rescued by constitutive HMP-1 activation. Indeed, RNAi against *afd-1* reduced total HMP-1/α-catenin accumulation at the vertex, as well as the spatial extent of HMP-1/α-catenin accumulation at the vertex in the case of both salt bridge mutated ("extended") and wild-type HMP-1 (Fig 6C and 6D). Taken together, these results suggest that whereas SRGP-1 may play an important role in orienting cells during rosette formation and organizing HMP-1/α-catenin around the vertex, AFD-1 is essential for general maintenance of HMP-1/α-catenin at the vertex following the final internalization events of gastrulation. Unfortunately, we could not perform the converse experiment to address whether AFD-1 recruitment requires HMP-1/α-catenin at this stage of development, because depletion of maternal and zygotic HMP-1/α-catenin leads to catastrophic morphogenetic failure, including lack of cleft closure [31,83].

We also attempted to assess whether introducing salt bridge mutations into HMP-1 positively affects AFD-1 accumulation and whether AFD-1/afadin accumulation depends critically on SRGP-1/srGAP. While there was an increase in AFD-1 accumulation in *hmp-1$^{R551/554A}$* homozygotes, it did not quite rise to statistical significance (S4 Fig). Based on this result, we conclude that AFD-1 recruitment is only mildly sensitive to the activation state of HMP-1/α-catenin. Moreover, since loss of AFD-1 strongly synergizes with weak loss of hmp-1 function [65]; present results), AFD-1 may be largely recruited to rosettes via a HMP-1/α-catenin-independent mechanism. This inference is consistent with data reported elsewhere indicating that AFD-1 recruitment is sensitive to loss of HMR-1/cadherin but not loss of HMP-1/α-catenin during Ea/p internalization [82]. We did not find a significant change in AFD-1 accumulation in *srgp-1$^{W122Stop}$* mutants (S4 Fig). Based on this result, and since double loss of function for *srgp-1* and *afd-1* leads to synergistic cleft closure defects (see Fig 5D), we favor the interpretation that the two proteins act largely independently during cleft closure and sealing.

## Discussion

### Rosette formation during C. elegans gastrulation requires cadherin-based adhesion

Ventral cleft closure is the culmination of gastrulation in the *C. elegans* embryo. It is essential for proper organization and cohesion of neuroblasts following gastrulation, which in turn is

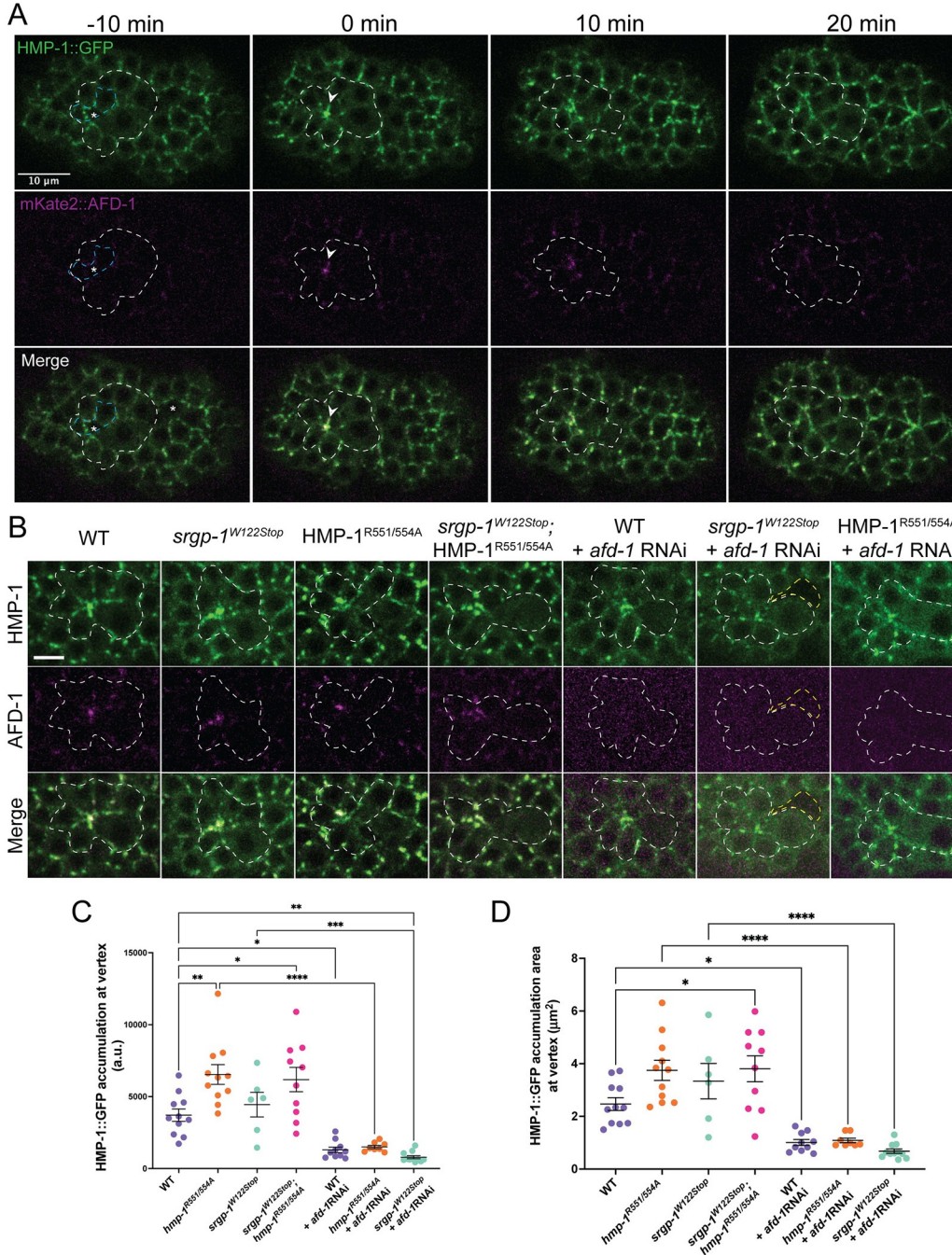

**Fig 6. AFD-1 accumulates at the vertex of the anterior rosette. (A)** A typical embryo expressing HMP-1::GFP and mKate2::AFD-1 before, during, and after anterior rosette formation. White dotted line outlines cells that form the rosette. Blue dotted lines mark the anterior cells that internalize prior to formation of the rosette. White arrowheads indicate the vertex of the rosette. Scale bar is 10 μm. **(B)** Images of HMP-1::GFP and mKate2::AFD-1 localization at the anterior rosette immediately following internalization in various genetic backgrounds. White dotted lines outline cells forming the rosette; yellow dotted lines indicate ventral cleft that remains open. Scale bar is 5 μm. **(C)** A graph depicting the total accumulation of HMP-1::GFP at the anterior rosette. (D) Graph depicting the area of HMP-1::GFP aggregation at the vertex of the anterior rosette. ****, p < 0.0001; ***, p<0.001; **, p<0.01; *, p < 0.05.

crucial for the embryo to survive the mechanical forces that operate during later morphogenesis [5,9,12]. Here we have characterized the cell rearrangements that accompany sealing of the ventral surface of the embryo, as cells on the surface change position to accommodate loss of cells that internalize near the end of gastrulation. Specifically, we have demonstrated that HMP-1/α-catenin and two of its functional modulators, SRGP-1/srGAP and AFD-1/afadin, facilitate the adhesion of cells during this critical stage of embryogenesis in the *C. elegans* embryo. Based on prior work, rosettes that form as a result of earlier cell internalization events in *C. elegans* appear similar [12], so insights gleaned from studying these later events will likely be useful in understanding other internalization events in the earlier embryo.

Cell internalization is a common event during gastrulation in metazoan embryos, as cells destined for the embryo's interior detach their apical surfaces from the embryo's exterior [2–5]. Given the apical-to-basal axis of such movements during *C. elegans* gastrulation, internalization also bears similarities to other basal extrusion events, often triggered by apoptosis or cell crowding in a variety of epithelia (reviewed in [84–86]). In all these cases, however, relatively little attention has been paid to how the cells that remain on the surface seal breaches on the embryonic exterior left behind by internalizing cells. At least in some cases, such tissue sealing involves multicellular rosette formation. The geometry of these rosettes bears similarities to those associated with other morphogenetic processes, such as convergent extension [53]. An intriguing parallel to the rosettes we observed are those observed in the chick epiblast [3,87,88]. Although the functional significance of the rosette structures in the primitive streak is unclear, these may reflect similar events at sites where cells depart from the surface of the embryo during gastrulation.

Rosettes in other contexts, such as during convergent extension in the *Drosophila* germband, involve modulation of adhesion complexes as cells change their connections to one another [52,89]. In contrast, little is known about adhesive changes among neighboring non-intercalating cells that seal gaps left behind by ingressing cells. In the case of sea urchin primary mesenchyme cells, which exhibit many aspects of standard epithelial-mesenchymal transition [90], cells lose cadherin-catenin complex components at the time of ingression [90,91]. The situation may be different in gastrulating cells in *C. elegans* and *Drosophila* neuroblasts; in the former, at least in the case of internalization of endodermal founder cells Ea and Ep, CCC components are transiently upregulated during apical constriction [11], while in the latter, post-translational loss of CCC components can be uncoupled from ingression events [92]. These differences indicate that while many processes may be conserved during internalization events, there may be a variety of mechanisms involved.

Our results shed light on this relatively understudied process by demonstrating that rosette formation at the end of gastrulation in *C. elegans* requires a robust cell-cell adhesion machinery. Reduction in the ability of HMP-1/α-catenin to bind F-actin in $hmp\text{-}1^{S823F}$ mutants leads to an increase in ventral cleft closure failure, as does loss of the HMP-1 binding partner, SRGP-1. The cells surrounding the position of the vacated cell at the end of ventral cleft closure form a rosette, which ultimately resolves as cells make new connections to one another at the site of sealing.

## Rosette formation is fostered by activating HMP-1/α-catenin

We and others have shown that the α-catenin M domain engages in interactions that regulate the C-terminal F-actin binding region of α-catenins [42,93]. In this context it is striking that the $hmp\text{-}1^{R551/554A}$ mutation suppresses phenotypes associated with the *S823F* mutation, which we have shown previously measurably decreases the F-actin binding activity of HMP-1/α-catenin [94]. Previous intragenic suppressors all clustered in the C terminus of HMP-1/α-

catenin, not in the M domain [94]. Our present results provide further evidence that the conformation of the M domain is relevant to the ability of HMP-1/α-catenin to interact, either directly or indirectly, with the actin cytoskeleton.

## Rosette formation depends on proper HMP-1/α-catenin localization mediated by both the C-terminus and F-BAR domains of SRGP-1/srGAP

Salt bridges in the M domain of mammalian αE-catenin stabilize the M domain in a "closed" conformation, reducing the likelihood of association of vinculin [58,67]. In *C. elegans*, however, we have shown previously that DEB-1/vinculin is confined to myoblasts in the early embryo, and that it does not bind HMP-1/α-catenin [55,94], suggesting that HMP-1/α-catenin interacts with other effectors in non-muscle cells. In addition to its utility in identifying intramolecular interactions that regulate HMP-1/α-catenin activity, the *hmp-1$^{S823F}$* mutation has been useful as a sensitized background for identifying such functional interactors. Both SRGP-1/srGAP and AFD-1/afadin were identified in a genome-wide RNAi screen for such interactors [65]. Our previous analysis indicated that the C terminus of SRGP-1 can physically bind the HMP-1/α-catenin M domain, but, unlike the case with vertebrate αE-catenin and vinculin, not when the HMP-1/α-catenin M domain is fully extended. We also showed that both the N-terminal F-BAR and C-terminal domains of SRGP-1 are functionally important during elongation [35].

Our analysis here also revealed roles for the N- and C-terminal regions of SRGP-1/srGAP during ventral cleft closure (visually summarized in Fig 7A). HMP-1/α-catenin becomes highly concentrated at the tips of cells at rosette vertices in embryos expressing endogenously tagged, full-length SRGP-1. The greater severity of gross morphological defects in *srgp-1* nonsense and C-terminal deletion mutants further suggests a more stringent requirement for the C terminus, which is lacking in both mutants, in stabilizing HMP-1/α-catenin. Since the SRGP-1 C terminus is intact in *srgp-1$^{ΔF-BAR}$* mutants, it is possible that, whereas SRGP-1$^{ΔF-BAR}$ can no longer interact with the membrane directly to stabilize the CCC, when the N-terminus is absent SRGP-1 can still interact with HMP-1/α-catenin in some functional capacity. In this case HMP-1/α-catenin presumably exclusively relies on its association with the heterotrimeric cadherin/catenin complex to associate with the plasma membrane, which is less efficient in recruiting HMP-1/α-catenin at sites of high membrane curvature, such as cell tips at rosette vertices. The likelihood of this possibility is strengthened by our observation that *srgp-1$^{ΔF-BAR}$* homozygous embryos exhibit less lethality than *srgp-1$^{W122Stop}$* and *srgp-1$^{ΔC}$* homozygotes. In our previous work we suggested that there may be a second region of SRGP-1, which lies N-terminal to the C-terminal region, that can interact with some junctional component—possibly including HMP-1/α-catenin [28,35]; our present work is consistent with this possibility. A distinct role for the SRGP-1/srGAP N terminus is also suggested by our results. When the SRGP-1 F-BAR domain is deleted the tips of cells in the rosette are blunted (see Fig 3A vs. B at 5 min) and HMP-1/α-catenin forms multiple aggregates in cells in the rosette, corresponding to less robust rosettes. SRGP-1 may either stabilize highly curved regions of the plasma membrane at rosettes or be recruited to such sites, thereby stabilizing HMP-1/α-catenin. Future experiments will be needed to distinguish these possibilities.

The *hmp-1$^{R551/554A}$* mutation, which maintains the HMP-1/α-catenin M domain in an open conformation [35], can suppress cleft closure defects caused by loss of the SRGP-1/srGAP C-terminus, but not those resulting from loss of the SRGP-1 F-BAR domain (see Fig 4B). There are several potential explanations for this result. One possibility is that the C terminus of SRGP-1 regulates HMP-1/α-catenin function beyond localization. For example, the C terminus of SRGP-1, once bound, could facilitate further opening and activation of the HMP-

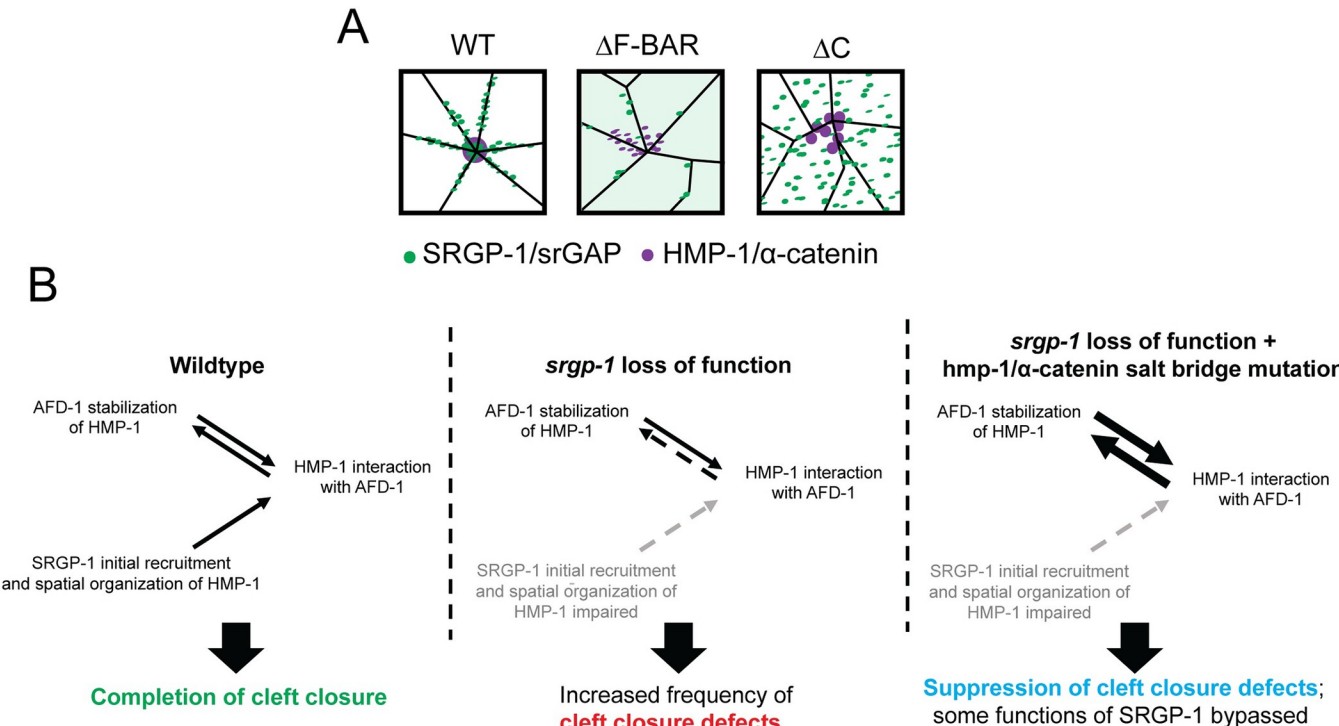

**Fig 7. Summary of the roles of SRGP-1/srGAP and AFD-1/afadin in stabilizing HMP-1/α-catenin at rosettes. (A)** Visual summary of HMP-1 and SRGP-1 distribution in cells in different *srgp-1* mutant backgrounds. **(B)** Summary of proposed routes by which HMP-1 is stabilized in wild-type and *srgp-1^{W122Stop}*; *hmp-1^{R551/554A}* backgrounds. (Left) In wild-type embryos, SRGP-1 and AFD-1 function in different ways to modulate HMP-1 during cleft closure. SRGP-1 may aid rapid recruitment and spatial organization of HMP-1 at sites of high membrane curvature in rosettes. AFD-1 may be recruited to cell-cell contacts at the ventral cleft independently of HMP-1. Once there, however, it may reciprocally stabilize HMP-1 at cell-cell contacts (two-way arrows). (Middle) When SRGP-1 function is impaired, initial recruitement and/or spatial organization at rosettes is impaired. This leads to reduced interactions of HMP-1 and AFD-1 (dashed arrow). (Right) When HMP-1 is activated by introducing salt bridge mutations it may be able to bypass SRGP-1 requirements, possibly via enhanced interaction with AFD-1 (bold two-way arrows).

1/α-catenin M domain, leading to recruitment of other binding partners. Constitutive opening of the HMP-1/α-catenin M domain in *hmp-1^{R551/554A}* mutants could obviate this requirement. Alternatively, if the major role of the SRGP-1 C terminus is to fine-tune the spatial organization and/or to increase the efficiency of localization of HMP-1, the enhanced activity of a fully open HMP-1/α-catenin could offset the quantitative loss of HMP-1 at nascent junctions in rosettes.

## Rosette formation is fostered by recruitment of AFD-1/afadin

Our experiments using a salt bridge mutant form of HMP-1/α-catenin led us to seek additional proteins that foster cell-cell adhesion during ventral cleft closure, leading us to AFD-1/afadin. AFD-1 accumulation at the vertex of the anterior rosette is striking compared to the low levels of AFD-1 accumulation elsewhere at this stage of development, including the posterior rosette, within which we did not see a similar accumulation of AFD-1. This indicates that junctions at the anterior rosette vertex are unique among cell-cell adhesions between ventral cells in the embryo at this stage, and that AFD-1/afadin is crucial for stabilizing them. Vertebrate afadin and *Drosophila* Canoe are recruited to junctions under increased tension or to sites with increased cellular (and hence actomyosin) dynamics [95–97]. Work that appeared while this work was in review further suggests that the *Drosophila* α-catenin M domain is important in coordinating afadin function [43].

In *C. elegans*, during later development when the epidermis is under substantial mechanical tension, AFD-1 appears at epidermal cell-cell junctions [65], consistent with tension-induced recruitment at that stage. That AFD-1 is recruited to the tips of cells in the anterior rosette vertex at the end of gastrulation suggests that the tips of these cells likewise experience increased tension. As the internalization event that forms the anterior rosette concludes, multiple cells must converge to create new contact points, which are susceptible to mechanical failure. A similar accumulation of AFD-1 is not observed in the posterior rosette, however. Notably, the cells of the posterior rosette undergo rapid extension shortly after rosette formation, whereas the anterior rosette persists. Work in *Drosophila* has demonstrated a necessity for Canoe localization to maintain tricellular junctions experiencing high tension; however, prolonged and continued accumulation of Canoe at junctions prevents vertex resolution during cell rearrangement [80]. If AFD-1 works in a similar fashion in *C. elegans*, this could imply that AFD-1 is required to stabilize the anterior rosette under higher mechanical loads, an idea that awaits further mechanobiological experiments.

We also found that, as is the case for *srgp-1*, loss of *afd-1* function leads to ventral cleft closure defects that can be suppressed via the *hmp-1*$^{R551/554A}$ mutation. Moreover, simultaneous depletion of SRGP-1 and AFD-1 leads to synergistic ventral cleft closure defects. One reasonable model that accounts for this data is that, while SRGP-1 fosters the initial recruitment of HMP-1 to nascent contact sites within rosettes, AFD-1 subsequently stabilizes more mature adhesions, allowing them to withstand tension prior to rosette resolution (Fig 7B). In this case, forcing HMP-1 into an open conformation may be able to bypass functional requirements for SRGP-1 by increasing the stability of adhesions through additional AFD-1 recruitment. It remains unclear whether AFD-1 can directly interact with HMP-1 in the way that their vertebrate counterparts do [59,73], or if AFD-1 is recruited to cell-cell adhesion sites through other effectors that in turn depend on an open conformation of HMP-1. Since defects in *afd-1* (RNAi); *srpg-1*$^{W122Stop}$ double loss-of-function embryos are still suppressed by *hmp-1*$^{R551/554A}$, there may be additional mechanisms that are stimulated by an open conformation of the HMP-1 M domain.

## Cell sealing at the end of gastrulation may involve multiple mechanisms

The rupture phenotype we observed in various *srgp-1/srGAP* and *afd-1/afadin* loss of function backgrounds could arise due to one or more non-mutually exclusive cellular mechanisms. Sealing failure could result from failure of surface cells to make new connections with one another, but rosette formation and sealing could also be fostered via maintenance of cadherin-dependent connections between surface cells and internalizing cells. Such adhesion would presumably be tuned precisely so that internalization draws cells inward while promoting midline movement of cells remaining on the surface that are attached to the remaining stalk connecting internalizing cells to the surface. Unfortunately, current technology does not allow us to perform clean experiments to test this possibility, which would require rescue of *srgp-1* function in internalizing cells vs. those remaining on the surface. In addition, the lability of fluorescent signals of tagged proteins in our imaging experiments did not permit the highly temporally and spatially resolved 4d imaging needed to adjudicate these questions. Future experiments will be required to further tease apart the roles of SRGP-1 in coupling surface cells to internalizing cells as opposed to cell-cell adhesion among surface cells at the end of gastrulation.

In conclusion, this work has clarified how cadherin-dependent adhesion between non-internalizing neighbors of internalizing cells, supported by SRGP-1/srGAP and AFD-1/afadin, stabilizes nascent cell-cell adhesions following the internalization events of gastrulation. Future

work focused on identifying other factors that play a role in anterior rosette formation and that dissects the mechanisms through which SRGP-1, AFD-1, and HMP-1 work together in this process should continue to clarify the cellular events of tissue sealing following internalization.

## Materials and methods

### Strains and genetics

*C. elegans* were maintained using standard methods. Bristol N2 was used as wildtype. A complete list of strains and genotypes used in this manuscript can be found in S1 Table.

### DIC imaging

Four dimensional DIC movies were collected on either a Nikon Optiphot-2 microscope connected to a QiCAM camera (QImaging) or an Olympus BX50 microscope connected to a Scion CFW-1512M camera (Scion Corp.) using Micro-Manager software (v. 1.42) [98,99]. ImageJ plugins (https://worms.zoology.wisc.edu/research/4d/4d.html) were used to compress and view DIC movies. All embryos were mounted on 10% agar pads in M9 solution unless otherwise specified.

### Confocal imaging

Embryos were dissected from adult hermaphrodites and mounted onto 10% agar pads in M9 solution and imaged. Movies were acquired over a period of 60 minutes at 1-minute intervals. For fluorescence imaging, a Dragonfly 500 spinning disc confocal microscope (Andor Corp.), mounted on a Leica DMi8 microscope, equipped with an iXon-EMCCD camera and controlled by Fusion software (Andor Corp.) was used to collect images using 0.21 μm slices with a 63×/1.3 NA glycerol Leica objective at 20°C.

### CRISPR/Cas9 genome editing

All novel knock-in and deletion alleles with *jc##* designation were generated via plasmid-based CRISPR/Cas9 [100] using repair templates cloned by SapTrap cloning [101]. Small substitution mutations were made via marker-free genome editing [102]. Guides, homology arm primers, and single-stranded repair templates for all CRISPR/Cas9 editing can be found in S2 Table.

### Injection RNAi

Injection RNAi was performed by synthesizing double-stranded RNA (dsRNA) using a T7 Megascript kit (Invitrogen). The templates for *srgp-1* and control RNAi were obtained from a feeding library [103]. pIC386 was used as a template for production of *afd-1* dsRNA. dsRNA was injected at a concentration of 2μg/μL in nuclease free water. L4 worms were injected and aged overnight before embryos were dissected from mature adults for imaging.

### Quantification and analysis

Percentage cleft closure defects were measured from embryos mounted for DIC imaging. Embryonic lethality was quantified by dividing the number of unhatched embryos laid on a plate by the total number of embryos on the plate from a single hermaphrodite. Total accumulation (integrated signal) and aggregation size for HMP-1::GFP were measured by drawing a circle around GFP signal at the vertex immediately following the internalization event.

## Statistical analysis

Data from control and experimental groups were compared using one-way ANOVA with Tukey post hoc testing to assess significance between individual groups. Contingency table analysis of HMP-1/α-catenin distribution was performed using Chi-square analysis. All statistical analyses were carried out in Prism (GraphPad Corp.).

## Supporting information

**S1 Fig. Categorization of HMP-1/α-catenin localization defects in wildtype and *srgp-1/srGAP* mutants.** (A) Visual summary of the three basic patterns of SRGP-1 and HMP-1 localization based on Fig 3. (B) Incidence of HMP-1 localization defects in wildtype, *srgp-1$^{\Delta F\text{-}BAR}$*, and *srgp-1$^{\Delta C}$* mutants. ***, $p < 0.001$ (Chi-square analysis).
(TIF)

**S2 Fig. *hmp-1$^{R551/554A;S823F}$* embryos lose their ability to suppress actin binding defects when exposed to greater compression.** Embryonic lethality of *hmp-1$^{R551/554A}$*, *hmp-1$^{S823F}$*, and *hmp-1$^{R551/554A;S823F}$* homozygotes when mounted on pads of 5% or 10% agar.
(TIF)

**S3 Fig. Depletion of *srgp-1* or *afd-1* suppress defects in a putative actin-binding gain-of-function allele of *hmp-1*.** (A) A domain map of HMP-1 showing the location of the QNLM676-679GSGS mutation. (B) A MUSCLE alignment of *C. elegans*, human, mouse, and *Drosophila* α-catenin at the RAIM site that was converted to GSGS by Ishiyama et al. (2018). Residues that have similar properties are assigned the same color. (C) Embryonic lethality on DIC mounts. (D) Percentage of cleft closure defects observed on DIC mounts. ****, $p < 0.0001$; *, $p < 0.05$.
(TIF)

**S4 Fig. mKate2::AFD-1 accumulation at the vertex of the anterior rosette is increased in *hmp-1$^{R551/554A}$* mutant backgrounds.** A graph depicting total accumulation of mKate2::AFD-1 at the vertex of the anterior rosette. *hmp-1$^{R551/554A}$* mutant backgrounds show an increase of AFD-1 accumulation, however none of the differences between groups rise to the level of statistical significance. p-values are indicated.
(TIF)

**S1 Table. Strains used in this study.**
(DOCX)

**S2 Table. Oligonucleotides used in CRISPR/Cas9 generation of alleles.**
(DOCX)

**S3 Table. Matrix of p-values for all pairwise comparisons in Fig 5D.**
(DOCX)

**S1 Data. Excel spreadsheet containing raw data, organized by figure.**
(XLSX)

## Acknowledgments

Some strains were provided by the *C. elegans* Genetics Center, which is funded by the NIH Office of Research Infrastructure Programs (P40 OD010440).

## Author Contributions

**Conceptualization:** Joel M. Serre, Jeff Hardin.

**Formal analysis:** Joel M. Serre, Jeff Hardin.

**Funding acquisition:** Bob Goldstein, Jeff Hardin.

**Methodology:** Joel M. Serre.

**Project administration:** Jeff Hardin.

**Resources:** Mark M. Slabodnick.

**Supervision:** Jeff Hardin.

**Validation:** Joel M. Serre.

**Visualization:** Joel M. Serre.

**Writing – original draft:** Joel M. Serre.

**Writing – review & editing:** Joel M. Serre, Bob Goldstein, Jeff Hardin.

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
