## [Decision Letter · Decision Letter 0]

4 Dec 2022

Dear Dr Hardin,

Thank you very much for submitting your Research Article entitled 'SRGP-1/srGAP and AFD-1/Afadin stabilize HMP-1/⍺-Catenin at rosettes to seal internalization sites following gastrulation in C. elegans' to PLOS Genetics.

The manuscript was fully evaluated at the editorial level and by independent peer reviewers. The reviewers appreciated the attention to an important topic, but identified some concerns that we ask you address in a revised manuscript. Also, we note that individual determinations underlying results presented in panels within Figures 1, 4 and 5, as well as Figures S1, are not available in a supplementary file or at a permanent DOI; the PLOS Genetics data availability policy (below) requires that manuscripts report individual determinations that gave rise to the results. Additionally, it would be appropriate to explicitly state in the Methods section if scientists scoring cleft closure or HMP-1::GFP were blinded as to genotype/treatment. 

We therefore ask you to modify the manuscript according to these and the review recommendations. Your revisions should address the specific points made by each reviewer or raised in the preceding paragraph.

Yours sincerely,

Anne C. Hart

Academic Editor

PLOS Genetics

Gregory Barsh

Editor-in-Chief

PLOS Genetics

Reviewer's Responses to Questions

**Comments to the Authors:**

Reviewer #1: Serre et al examine how adhesion is regulated at new cell-cell contact in the context of C. elegans gastrulation, with insights on HMP-1/⍺-catenin and its modulators SRGP-1/srGAP and AFD-1/Afadin. At the cellular level, the results show that srgp-1 loss of function result in a different spatial configuration of cells surrounding the internalizing cell when they converge to cover the internalizing cell, indicating an important role of SRGP-1 in sustaining tension during the process. At the molecular level, the results provide new insights such as function of the SRGP-1 N terminal domain, and AFD-1 acting through the open conformation of HMP-1. The study involves multiple CRISPR-edit alleles in order to gain the detailed insights in a complex in vivo process, as well as careful sorting of phenotypes in an extensive set of genetic conditions. Some more careful discussion and/or additional data analysis if practical, would strengthen the manuscript.

1. The cellular phenotype of srgp-1. The formation of "regular" cell contact instead of rosette could be resulted from less effective coupling between the internalizing cells and the surrounding cells and therefore less efficient extension of the surrounding cells. The authors discussed the role of SRGP-1 in sustaining curved membrane -- such curved membrane/tip not only exist in a rosette but also prior to that (more obvious in earlier gastrulating cells when cells are bigger). In this case, the phenotype would provide insights on the adhesion with the internalizing cells, and enrich the "adhesion clutch" model for internalization. A phenotype that could be attributed purely to the new contact between the surrounding cells would be the torn-up/burst to expose the internalized cells after complete sealing. The Methods section did not mention the temporal resolution of the imaging, but if it is higher than the 5 to 10 minutes shown in the figures, it may be worth analyzing the phenotypes before complete sealing as well as to examine if there is torn-up after complete sealing.

2. The cellular phenotypes of afd-1 loss of function. The manuscript offered the frequency of ventral closure defect, but did not provide the more detailed cellular phenotypes as in srgp-1. There may be some missed opportunity here: AFD-1 may function in the adhesion/coupling between the internalizing cells and the surrounding cells during internalization. The recent study in Drosophila germband extension suggests that Afadin/Canoe is recruited to rosette centers to cope with extremely high tension. Should AFD-1 function during internalization in addition to being at the rosette center, it would an important insight.

3. Regarding AFD-1 at rosette centers, the authors appear to favor the interpretation by the Drosophila study. Some of the experiments in the manuscript could provide a test. The cellular phenotypes of srgp-1 ("regular" cell contact) removes the extreme curvature associated with rosette centers. One would predict that AFD-1 localization would be reduced in these cases? Figure 6B examines AFD-1 localization in srgp-1 loss of function. However, the example image shows the least severe phenotype of srgp-1 where the rosette is there albeit with a less focused center.

4. The authors provide a nice discussion of related morphogenesis processes, such as basal extrusion and multicellular rosettes in convergent extension. It would be helpful to point out more explicitly that the multicellular rosettes in worm gastrulation involves a fundamentally different process (apical contraction in internalizing cells) than the multicellular rosettes in convergent extension (driven by edge contraction within a contiguous epithelium). The fact that the two mechanisms converge to the similar geometry of rosettes means that they could have shared aspects in terms of tension and adhesion. Making such a distinction would enhance the value of worm gastrulation as a model and thereby the value of the manuscript. Furthermore, the convergent extension type of rosettes also exist in the worm (ref 29 cited in the manuscript, convergent extension of the VNC). Making the distinction would help people be aware of the differences (and the complexity of worm morphogenesis). Finally, the authors mentioned rosettes in chick primitive streak. Some people attribute rosettes in vertebrate gastrulation to convergent extension. If the cited rosettes in chick is associated with cell internalization, it will be valuable to point that out and argue that these are worm gastrulation type of rosettes, and not the convergent extension type.

Reviewer #2: The manuscript by Serre et al. shows that anterior and posterior rosettes form after gastrulation to re-arrange cells and close a ventral cleft formed after cell internalization. They show that Ealpha-catenin/HMP-1, SRGP-1 and AFD-1 are enriched at the apex of the anterior and/or posterior rosettes, and propose a mechanism where SRGP-1 and AFD-1 help stabilize HMP-1 for adhesion, and to control the apical curvature of cells forming the vertex. Failure to properly stabilize HMP-1 at these rosettes could prevent proper cell rearrangement, causing a failure in cleft closure and subsequent morphogenetic defects. The manuscript is well-written, clear and accessible, although I do have some minor recommendations to improve accessibility to a broader (non-C. elegans) audience. The figures are nicely presented, and include robust statistical analyses. The authors are careful to propose a more simplistic model that is supported by their data, which is appreciated. However, they could consider performing an additional experiment to expand their model.

Minor comments

Non-C. elegans readers will find it challenging to follow the results based on the C. elegans-specific gene names, which are not always indicative of their function. For example, hmp-1 encodes HMP-1, which is the Ealpha-catenin homologue. Although they need to use gene names where appropriate, it would be helpful to refer to the protein in a more accessible way (e.g., HMP-1/Ce alpha-catenin).

The timing of gastrulation and internalization of Ea/Ep cells in relation to the cleft and the rosettes is not clear for the reader. For example, are the cells that are moving in while the rosettes form also part of the Ea/Ep internalization, or are these cells moving in after as part of the cell re-arrangement to close the cleft? All of the movies from Figure 2 onward have time=0 min when rosettes are clearly visible at the ventral surface. However, Figure 1 has time=0 min as ‘before last ingression’. Are these the same thing? What is ‘before last ingression’? If it means something else (the word before is confusing), they should be consistent with timing between figures. Also, adding a schematic to show the relative timing of all of these events would be helpful (e.g., internalization of Ea/Ep, cleft closure, rosette formation and resolution). Especially since we are only shown DIC movies for cleft closure phenotypes, but then are only shown fluorescence images for rosette phenotypes, this makes it confusing to interpret the timing of cleft closure defects in the mutants.

As a related comment, the rosettes appear to be quite large, covering a large portion of the ventral surface, while the cleft is relatively small. It is hard to see how minor perturbations in the rosettes (e.g., stability or curvature of cells) could cause such drastic effects on cleft closure. Are there adhesion defects between other 'non-rosette' ventral cells, or is this more strictly because of mechanical forces causing rupture?

Is the w122stop allele a null? It would be good to clarify this for the reader.

Lines 135-140 are not clear as written. The authors indicate that there are cleft closure defects in embryos with the delta F-bar mutant allele, but then say that they are not significant (and is ns on their graph). They could change the wording to clarify that this means something given that there are no defects in control embryos. I imagine this is an issue with n’s.

The flow from the section describing Figure 3 to Figure 4 could be improved to better explain the rationale for those experiments. One way to do this is to better connect to the previous finding. For example, the authors show that sgrp-1 alleles are differentially required for hmp-1 localization, so mutating hmp-1 to make protein that is in an extended conformation could be predicted to override the requirement for srgp-1.

Lines 183-184 are confusing as written. “.. as a result, the HMP-1 M domain adopts a constitutively open conformation that prevents the recruitment of the SRGP-1 C-terminus..” Why? I think the issue is that there is no description of the structural function of alpha-catenin (or HMP-1) in the introduction, so the reader is learning this information for the first time in the results. Similarly, the follow-up statements in lines 185-187 where they refer to how an extended conformation of alpha-catenin activates actin binding or recruitment of vinculin and afadin. How does this relate to HMP-1 structurally? I think it is an issue where information on the structural regulation of alpha-catenin needs to be provided to the reader earlier in the manuscript (introduction) to understand for the rationale behind making the salt-bridge mutants. They could keep a sentence in the results to remind the reader of this relevant information.

Figure 4. Since the delta Fbar phenotypes are already weaker, it might be challenging to detect suppression with the double/triple mutants. I think it is an issue of being cautious of interpretation. For example, saying that the C-term mutant suppresses is fine, but I would say it is inconclusive for the delta Fbar mutant.

Figure 4C. The increase in egg-laying with the salt-bridge mutant is really interesting, and supports a change in mechanical properties in the spermatheca – e.g., actomyosin contractility, where srgp-1 is known to function. This is really interesting, and the authors could comment on this finding in the discussion.

Lines 233-235. Again, more knowledge of the structural regulation of alpha-catenin and how it interacts with other regulators of F-actin and adhesion would be helpful to have in the introduction. It seems that the role of the M domain in regulating actin-binding is important, but is not described.

Figure 5B – The y-axis is called 'embryonic arrest' on the graph, but there is a ‘hatch’ category, which doesn’t make sense. Should this be % phenotype or another term that encompasses the different categories?

Figure 5B, C – To help obtain more support for their model, did the authors try a double knockdown of both srgp-1 and afd-1 in the hmp-1 mutant? Or is the phenotype already at ‘max threshold’ with srgp-1 and hmp-1?

Figure 5D – It is not clear why the authors did not also test the knockdown of both srgp-1 and afd-1 in the hmp-1 strain with both hypomorphic and salt-bridge mutants. Especially if they want to show that the suppression is alleviated by the loss of afd-1. I appreciate that the lethality increases with both knockdowns and the hmp-1 allele with the salt bridge mutants, but adding the hypomorphic mutation might give a clearer result.

Figure 6. To further address mechanism, the authors should consider looking at differences in F-actin (and myosin) localization/enrichment in/near the anterior vs posterior rosettes. They could also see how this localization changes when HMP-1 with the salt-bridge mutants is enriched. This could help address the mechanical aspects of their model. For example, it isn’t clear if the salt-bridge mutants directly lead to an increase in actin-binding/crosslinking. Have the authors considered reducing myosin contractility via mild let-502 RNAi to reduce tension, and see how this affects localization of the HMP-1 salt-bridge mutants and/or AFD-1 at the anterior rosette?

In the discussion, the authors refer to the stabilization of HMP-1 in the context of its role in adhesion, but given that there is also a mechanical aspect to the CCC, it would be good to know how these would work in concert. Especially since there is some in vitro data suggesting that alpha-catenin binds mutually exclusively to E-cadherin or actin.

In the discussion, there is an emphasis on the role of SRGP-1 in controlling the curvature of the membrane of the cells in the rosette, but this was not shown or quantified in the results. The cell segment software shows the change in cell position and shape in 2D, but this statement refers to a parameter that requires measurements in 3D. For example, I don’t see the ‘blunted ends’ that the authors refer to in the srgp-1 mutant. To support their interpretation, they should show some 3D rendering and/or quantify differences. Or they could decrease this aspect of their discussion.

Reviewer #3: Summary

These studies extend previous work by these groups to understand the molecular and cellular mechanisms required to seal openings after embryonic gastrulation in C. elegans. This process is critical for preventing herniation along and the ordering of germ layers. Most of the genes analyzed have been shown to play roles at different stages of morphogenesis including epidermal enclosure, whereas this study focuses on an earlier event, closure of the ventral cleft, a prerequisite for epidermal enclosure and for positioning the ventral nerve cord.

Overall strengths: The studies generally appear rigorous and of high quality and address some interesting aspects of cell biology and morphogenesis. In particular, the experiments make excellent use of genome editing methods to generate specific types of mutations as well as endogenous reporters. This is coupled to extensive genetic analysis, in vivo imaging, and a good deal of phenotypic quantification. The introduction and discussion are detailed and scholarly.

Potential weaknesses: The rationale for performing certain studies was unclear. Likewise, some of the results were perplexing or hard to interpret. The above issues can presumably be addressed largely through writing. The overall low penetrance of phenotypes in some of the single mutant backgrounds is a minor cause for concern. Obviously, the data is the data and partially redundant mechanisms are likely the explanation. But it does make it more difficult to obtain statistically robust results when comparing various backgrounds given the natural variation present between embryos. Overall, the advance did seem somewhat incremental, but this is subjective.

Taken together, my overall recommendation is to accept pending what are likely to be fairly minor revisions.

Specific points by section

1) Fig 2. In legend correct what the white dashed line indicates (VC?) and change “White dotted lines outline… rosettes” to “Yellow dotted lined…”. Also, consider stating that HMP-1 also appears to accumulate to some extent at the posterior rosette vertex.

2) Fig 3. Reading through Fig 3 several points of minor confusion arose. (i) Fig 1 showed persistent clefts in the F-BAR and CT deletions, but these are not indicated/present in the embryos in Fig 3. (ii) Both mutants seem to show defects in the accumulation of HMP-1 at the vertex, but the text concludes that this a function of the CT. (iii) Likewise both mutants seem to have issues forming a vertex. (iv) Maybe include a statement about effect of F-BAR on SFGP-1 localization, as both mutants seem to affect this as well. (v) The phenotype of F-BAR was considerably weaker (ns) in Fig 1 but in Fig 3 both mutants seem to show similarly severe phenotypes. Obviously, this is going simply by what’s shown in Fig 3. So perhaps semi-quantitative statements in the text could be helpful such as, “Vertices failed coalesce in A out of B F-BAR mutants vs C out of D CT mutants.”

3) Fig S1. Might it make sense to include statistics for the comparison of hmp-1(S823F) vs hmp-1(R551/554A; S823F) – both 5% and 10% - to show that R551/554A is suppressing?

4) Fig 5. The interpretation of the genetic data wasn’t terribly straightforward and could use some clarification and interpretation. Along those lines, statistics for the comparisons of WT + afd vs hmp-1(R551/554A) + afd; srgp-1 + control vs srgp-1; hmp-1(R551/554A) + control; srgp-1 + afd vs srgp-1; hmp-1(R551/554A) + afd would be helpful/necessary. Without these comparisons statements such as “there may be additional factors beyond AFD-1 that may stabilize HMP-1 when it adopts an open conformation” may be hard to make. There is also the general conundrum is that hmp-1(R551/554A) seems to suppress in some cases (srgp-1 + control and srgp-1 + afd) and enhance in another (WT + afd). What does this mean exactly?

5) Fig S2. Again, most readers will need better clarification and interpretation of the genetic data, such as the finding that srgp-1 and afd show suppression effects for the hmp-1(QLNM) mutant whereas in Fig 5 with hmp-1(R551/554A), srgp-1 and afd were showing enhancement effects. Along those lines, cell biological data accompanies the genetic data for the various mutants except for the hmp-1(QLNM) mutant. Could this be informative?

6) Fig 6. (i) The specific questions being addressed by the experiments in Fig 6B were unclear. (ii) Fig 6/D were not referenced in the Results and the meaning/difference between C/D was not spelled out. (ii) The image of HMP-1 + afd looks very similar to WT, whereas the data in C/D suggest a major difference.

7) Fig S3. It’s stated that “we could not perform the converse experiment to address whether AFD-1 recruitment requires HMP-1 at this stage of development” but S3B indicates hmp-1(RNAi). Is this correct?

8) Discussion. The figure supporting models in the discussion (Fig 7) wasn’t hugely informative. Points brought up during the discussion could potentially more fleshed out in this figure, which would also help the reader to frame the data’s meaning and significance.

Minor

9) Update reference 35.

10) Consider changing “conferred” to “caused” on line 218.

**Have all data underlying the figures and results presented in the manuscript been provided?**

Reviewer #1: None

Reviewer #2: Yes

Reviewer #3: Yes

PLOS authors have the option to publish the peer review history of their article (what does this mean?). If published, this will include your full peer review and any attached files.

Reviewer #1: No

Reviewer #2: **Yes: **Alisa Piekny

Reviewer #3: No

---

## [Decision Letter · Decision Letter 1]

13 Feb 2023

Dear Dr Hardin,

We are pleased to inform you that your manuscript entitled "SRGP-1/srGAP and AFD-1/afadin stabilize HMP-1/⍺-catenin at rosettes to seal internalization sites following gastrulation in C. elegans" has been editorially accepted for publication in PLOS Genetics. Congratulations!

Yours sincerely,

Anne C. Hart

Academic Editor

PLOS Genetics

Gregory Barsh

Editor-in-Chief

PLOS Genetics

Comments from the reviewers (if applicable):

Reviewer's Responses to Questions

**Comments to the Authors:**

Reviewer #1: The authors have sufficiently addressed the issues. I do not have further concerns or comments.

Reviewer #2: The revised manuscript is well-written and accessible, and the authors have addressed all of my comments and concerns. This manuscript nicely shows how a genetic model system can be used to understand a developmental process at the cellular level. The authors show how rosettes form during gastrulation to reposition neuroblasts on the ventral surface after cell internalization. They strategically use different mutants to reveal a mechanism whereby depending on the conformation of HMP-1, the requirement for some complexes and/or components required to stabilize F-actin for adhesion could change. They also show that two distinct rosettes can form at the anterior vs. posterior of the embryo. Importantly, different molecular components, e.g., AFD-1, are recruited to these rosettes, which could reflect differences in mechanical tension, and influence how stable the rosette is. I think this study will be of interest to researchers in the cell and developmental biology community.

Reviewer #3: Nice job and thank you for addressing my comments.

**Have all data underlying the figures and results presented in the manuscript been provided?**

Reviewer #1: Yes

Reviewer #2: Yes

Reviewer #3: None

PLOS authors have the option to publish the peer review history of their article (what does this mean?). If published, this will include your full peer review and any attached files.

Reviewer #1: No

Reviewer #2: **Yes: **Alisa Piekny

Reviewer #3: No

**Data Deposition**

http://datadryad.org/submit?journalID=pgenetics&manu=PGENETICS-D-22-01268R1

**Press Queries**

---

## [Editor Report · Acceptance letter]

28 Feb 2023

PGENETICS-D-22-01268R1 

SRGP-1/srGAP and AFD-1/afadin stabilize HMP-1/⍺-catenin at rosettes to seal internalization sites following gastrulation in *C. elegans*

Dear Dr Hardin, 

We are pleased to inform you that your manuscript entitled "SRGP-1/srGAP and AFD-1/afadin stabilize HMP-1/⍺-catenin at rosettes to seal internalization sites following gastrulation in *C. elegans*" has been formally accepted for publication in PLOS Genetics! Your manuscript is now with our production department and you will be notified of the publication date in due course.

With kind regards,

Zsofia Freund

PLOS Genetics

On behalf of:
